# Clove Essential Oil and Its Main Constituent, Eugenol, as Potential Natural Antifungals against *Candida* spp. Alone or in Combination with Other Antimycotics Due to Synergistic Interactions

**DOI:** 10.3390/molecules28010215

**Published:** 2022-12-26

**Authors:** Anna Biernasiuk, Tomasz Baj, Anna Malm

**Affiliations:** 1Department of Pharmaceutical Microbiology, Faculty of Pharmacy, Medical University of Lublin, 20-093 Lublin, Poland; 2Department of Pharmacognosy with the Medicinal Plants Garden, Medical University of Lublin, 20-093 Lublin, Poland

**Keywords:** clove essential oil, eugenol, antifungal activity, *Candida* spp., mode of actions, interactions

## Abstract

The occurrence of candidiasis, including superficial infections, has recently increased dramatically, especially in immunocompromised patients. Their treatment is often ineffective due to the resistance of yeasts to antimycotics. Therefore, there is a need to search for new antifungals. The aim of this study was to determine the antifungal effect of clove essential oil (CEO) and eugenol (EUG) towards both reference and clinical *Candida* spp. strains isolated from the oral cavity of patients with hematological malignancies, and to investigate their mode of action and the interactions in combination with the selected antimycotics. These studies were performed using the broth microdilution method, tests with sorbitol and ergosterol, and a checkerboard technique, respectively. The CEO and EUG showed activity against all *Candida* strains with a minimal inhibitory concentration (MIC) in the range of 0.25–2 mg/mL. It was also found that both natural products bind to ergosterol in the yeast cell membrane. Moreover, the interactions between CEO and EUG with several antimycotics—cetylpyridinium chloride, chlorhexidine, silver nitrate and triclosan—showed synergistic or additive effects in combination, except nystatin. This study confirms that the studied compounds appear to be a very promising group of phytopharmaceuticals used topically in the treatment of superficial candidiasis. However, this requires further studies in vivo.

## 1. Introduction

In recent years, the occurrence of superficial infections caused by fungi belonging to *Candida* spp. has increased significantly, especially in immunocompromised patients (organ transplant recipients, cancer patients receiving chemotherapy, and people with HIV/AIDS) [1,2,3]. In these patients, fungi can also cause potentially life-threatening systemic disorders. The predominant cause of all types of candidiases is *Candida albicans*—simultaneously the fourth most common etiological factor of hospital-acquired infections. It is associated with mortality rates as high as 35–50% [1,2]. Other species of non-*albicans Candida* spp. (NAC), such as *C. glabrata, C. parapsilosis* or *C. krusei*, have also shown an increased incidence of nosocomial infections. However, superficial candidiasis, including oropharyngeal candidiasis, also create an important problem due to several high-risk populations in the community, e.g., oncological patients. Despite the high incidence and severity of *Candida* infections, treatments are still limited and insufficient. Currently, the list of the commercially available antifungal agents used for the treatment of infections caused by *Candida* spp. is limited to only a few classes: polyenes (e.g., amphotericin B or nystatin), azoles (e.g., fluconazole or posaconazole) and echinocandins (e.g., caspofungin or micafungin). Additionally, none of them meets all the expectations. The considerable number and intensity of side effects, mainly related to the usage of high antifungal concentrations, and the constantly increasing drug resistance (especially to azoles), makes fungal infections a serious clinical problem. Therefore, it is necessary to search for alternative therapies or novel antimycotics [1,2,3]. Recently, natural compounds with promising biological properties have been intensively investigated, e.g., essential oils (EOs) and their components. Using them to develop new potential phytopharmaceuticals would be an excellent idea.

Essential oils (EOs) have gained increased interest due to extensive biological activity. They are rich mixtures of chemical compounds belonging to different chemical families, including terpenes, terpenoids, aldehydes, phenols, alcohols, ethers, esters, ketones, and other aromatic and aliphatic constituents with low molecular weights [3]. Usually, the chemical characterization of many EOs reveals the presence of only 2–3 primary components at a fairly high concentration (20–70%) compared to other components present in trace amounts [3].

Clove EO (CEO) is obtained from spicy clove (*Syzygium aromaticum* Merr. Et Perry). The genus *Syzygium* (*Eugenia*) belongs to the Myrtaceae family. The homeland of cloves is the islands in Indonesia [4]. At present, approximately 100 compounds have been identified in various CEOs. The CEO’s dominant constituent is eugenol (EUG), which varies from 30–95%, with the lowest content of this compound (28%) in EO isolated from growing leaves. This EO also contains eugenol acetate, β-caryophyllene, α-ilangene, δ-cadinene, as well as compounds with an aromatic structure, i.e., methyl eugenol, anetol, chavikol, vanillin, benzyl alcohol, cinnamic aldehyde, benzyl salicylate, and calamenene [4,5]. The chemical composition of EOs depends on the origin of the oil and the degree of development of the leaves or pitch. Leaf oil differs from bud oil, with low content of eugenyl acetate. The Polish Pharmacopoeia defines the content of the following components in the CEO: eugenol (75–88%), β-caryophyllene (5–14%) and eugenol acetate (4–15%) [4].

Several reports have extensively documented the pharmacological effects of CEO and its main active constituent, EUG: antiviral [3,4,5,6], antibacterial [3,4,5,7,8], antifungal [3,5], anticancer [5,9], antioxidant [5,8], and anti-inflammatory activities [5]. These natural compounds are widely used as an antiseptic in oral diseases [3,4,10] and for the treatment of toothaches, allergy disorders, asthma, acne, scars, and rheumatoid arthritis, and they showed antispasmodic and acaricidal effects [11,12]. Moreover, the CEO has demonstrated aphrodisiac, antipyretic, appetizer, hypnotic, anxiolytic, antiemetic, analgesic, decongestant, antiepileptic, myorelaxant, and expectorant properties, as well as a medicinal influence against trophic disorder [13,14]. CEO and EUG are also responsible for the fungicidal effect against candidiasis, onychomycosis, and dermatophytosis [4,5,15,16,17,18,19].

In the present work, we first decided to verify the composition of the CEO. In the next stage, antifungal activity in vitro of CEO and its main constituent, EUG, against *Candida* spp. from immunocompromised patients was examined. In addition, the mechanism of anticandidal action on fungal cells and the effect of combination with selected antimycotics was investigated. 

## 2. Results

### 2.1. ATR-FTIR Fingerprint and Chemical Composition of CEO

Preliminary ATR-FTIR analysis (Figure 1) of CEO showed a similar vibration structure to EUG in the fingerprint region of 500–1500 cm^−1^. In the EO, stretching vibrations of the C=C aromatic bond (1511 cm^−1^) and asymmetric C-O-C bonds in the range of 1100–1265 cm^−1^, characteristic of EUG, were found. The occurrence of vibrations in the range of 1431–794 cm^−1^ may be caused by the presence of the CH_2_ group. In addition, vibrations at the wavelength of 1764 cm^−1^ were found in the EO, which may be derived from a carbonyl ester attached to the aromatic ring of eugenol acetate [8,20]. 

In the next stage of this study, the qualitative and quantitative analysis of EO components using the GC–MS method was assessed. The results are presented in Table 1 and Figure 2. The main chemical components in the CEO were EUG (66.81%) and β-caryophyllene (14.11%).

### 2.2. The Antifungal Activity Assessment of CEO and EUG

The obtained results showed the potential antifungal effect of CEO and EUG against two reference *Candida albicans* strains: *C. albicans* ATCC 2091 and *C. albicans* ATCC 10231, and three strains of non-*albicans Candida* spp. (NAC): *C. parapsilosis* ATCC 22019, *C. glabrata* ATCC 90030 and *C. krusei* ATCC 14243. Taking into account the data presented in Table 2, the activity of CEO and EUG was shown at MIC = 0.5–2 and 0.25–2 mg/mL, respectively. The MFC values were higher, in the range of 1–4 mg/mL. *C. albicans* ATCC 2091 was found to be the most susceptible to CEO (MIC = 0.5–1 mg/mL, MFC = 1–2 mg/mL). In turn, *C. krusei* ATCC 14243–to EUG (MIC = 0.25–1 mg/mL, MFC = 1–2 mg/mL). Moreover, both natural compounds showed fungicidal effects (MFC/MIC = 1–2) towards all reference *Candida* strains.

These data indicated the similar antifungal activity of both compounds. Therefore, this study aimed to confirm their effectiveness against clinical isolates of *Candida:* 30 isolates of *C. albicans* and 30 isolates of non-*albicans Candida* spp. (NAC) from the oral cavity of hospitalized patients with hematological malignancies, particularly vulnerable to infections, e.g., candidiasis. According to the distribution of MIC values among clinical isolates and data presented in Table 3 and Figure 3, the studied EO and its main constituent showed an antifungal effect towards strains with MIC = 0.25–2 mg/mL. The values of MFC were the same or 2–4-fold higher than MIC values.

The ranges of MFC were 0.5–4 mg/mL for all *Candida* spp. isolates, except NAC strains with MFC = 0.25–4 mg/mL. In addition, the MIC_50_ and MIC_90_ values were calculated, defined as the minimum concentrations inhibiting the growth of 50% or 90% of all tested strains, respectively. These values were: MIC_50_ = 0.5 mg/mL and MIC_90_ = 1 mg/mL against all studied *Candida* isolates (for CEO and EUG) and MIC_90_ = 2 mg/mL for NAC in the case of CEO. In turn, MFC_50_ and MFC_90_ values were described as the lowest concentrations required to kill 50% or 90% of the clinical isolates (MFC_50_ = 1 mg/mL and MFC_90_ = 2 mg/mL). 

As shown in Figure 3a, the largest number of *Candida* isolates was inhibited by CEO at a minimal concentration of 0.5 mg/mL – 18 (60%) and 17 (56.67%) *C. albicans* and NAC isolates, respectively. In the case of MFC, the concentrations were similar. Most of the *Candida* isolates—11 (36.7%) strains of *C. albicans* and 10 (33.3%) isolates of NAC—were killed at a minimal concentrations of 1 and 0.5 mg/mL, respectively. In turn, Figure 3b presents the frequency of MIC and MFC values for EUG. As with the CEO, the strains were mostly sensitive to this compound at MIC = 0.5 mg/mL (14 (46.7%) *C. albicans* isolates and 16 (53.3%) NAC strains) and MFC = 1 mg/mL (15 (60%) and 12 (40%) isolates of *C. albicans* and other *Candida* spp., respectively).

Moreover, the determination of the MIC and MFC values of the CEO and EUG allowed for the assessment of the MFC/MIC ratio (Table 4). On this basis, their fungicidal activity (MFC/MIC ≤ 4) was found against all the tested clinical isolates of *Candida* spp. No fungistatic effect (with values MFC/MIC > 4) was found. Analyzing the results in Table 4, it was observed that the most common ratio was 2 for both *C. albicans* and NAC isolates, with a frequency of 50–66.7%.

These data confirmed that CEO and EUG had a beneficial fungicidal effect towards all studied *Candida* spp. strains isolated from the oral cavity of patients with hematological malignancies.

### 2.3. Mode of Antifungal Action of CEO and EUG

The mechanism of action of CEO and EUG (as presented in Figure 4) was tested to determine whether its antifungal activity involves direct interaction with the cell wall structure of *Candida* spp. (via testing with sorbitol) and/or with the ion permeability of the membrane of this organism (via the test with ergosterol).

Sorbitol has an osmoprotectant function and is essential for fungal growth when fungi are in the presence of drugs that act on the cell wall. It is used to stabilize fungi protoplasts, protecting their cell wall from environmental stresses, particularly osmotic changes. In this assay, MIC determinations were conducted in parallel with and without 0.8 mol/L sorbitol. It is expected that the MIC of a compound that damages the cell wall will shift to a much higher value in the presence of the osmotic support. Results, as presented in Table 4, showed that MIC did not vary in the presence of sorbitol after 7 days of incubation for any of the yeasts tested, suggesting that both CEO and EUG would not act by inhibiting the mechanisms controlling cell wall synthesis. The MIC of nystatin used as a negative control, acting at the level of the fungal cell membrane, was also not altered in the presence of sorbitol.

The next step of this work was to determine if CEO and EUG act by affecting ergosterol in the fungal cell membrane. Ergosterol and enzymes of the ergosterol biosynthetic pathway are essential targets of several classes of antifungals used to treat *Candida* infections with a dominant position of the polyenes and the azoles. This test detects if a compound acts by binding to the ergosterol of the fungal membrane and is based on offering exogenous ergosterol to a compound which, when possessing affinity with it, will rapidly form a complex, thus preventing the complexation with the membrane’s ergosterol. As a consequence, an enhancement of MIC is observed. The obtained results, presented in Table 4, showed an increase in the MIC of both EO and its main compound in the medium with exogenous ergosterol compared with the ergosterol-free experiment. Their MICs values were 2–8-fold higher in the presence of ergosterol. In turn, an 8–16-fold increase in MIC was observed for the positive control drug nystatin, whose interaction with ergosterol is well known. These data suggest that CEO and EUG appear to bind to the ergosterol in the membrane, which increases ion permeability and ultimately results in cell death.

### 2.4. Investigation of Interaction of CEO and EUG with Selected Antimycotics

In the next stage of research, a combination of CEO and EUG with some antifungal drugs—nystatin and synthetic antiseptics—cetylpyridinium chloride monohydrate, chlorhexidine, silver nitrate and triclosan was also assayed for their effect on the growth of reference *Candida* spp. strains. The possible interactions of natural plant compounds with antifungals were determined using the checkerboard microtiter method. MICs of compounds alone, as well as MICs of combinations which exhibited inhibitory effects, were used to calculate fractional inhibitory concentrations (FICs) and Ʃ FIC (FICI – FIC index) values. To determine the interactions between the tested compounds and the chosen antifungal substances, FIC values were calculated according to the formula given by Blanco et al. [21]. In turn, based on the FIC value, the appropriate type of interaction was determined. 

The data shown in Table 5, Table 6, Table 7, Table 8, Table 9, Table 10, Table 11 and Table 12 (and Appendix A) for CEO and EUG in combination with antifungals indicate both synergistic, additive and indifferent interactions. No antagonistic effect towards *Candida* strains was observed. The MIC values of CEO and EUG alone were 1000 µg/mL for all strains. In turn, in the combination, their MICs were reduced to 2–16 fold depending on the compound and *Candida* strain. 

Both compounds showed synergy with cetylpyridinium chloride against all *Candida* spp. strains. FICI values were 0.375 against *C. parapsilosis* (for CEO) or *C. krusei* and *C. parapsilosis* (for EUG) and 0.5 towards other strains. MICs of CEO in combination decreased by 4–8 fold (MIC = 125–250 µg/mL) and EUG by 2–4 fold (MIC = 250–500 µg/mL) with FIC = 0.125–0.25. MICs of cetylpyridinium chloride were also reduced by 4–8 fold (from 0.48–3.91 to 0.12–0.98 µg/mL) depending on the strain.

The synergistic effect of CEO and EUG with chlorhexidine (FICI = 0.375–0.5) was also observed for all strains except *C*. *parapsilosis* (addition at FICI = 0.562). *Candida* spp. strains were 4–16-fold more sensitive to oil and its component in combination (MIC = 62.5–250 µg/mL and FIC = 0.125–0.5). MICs of this antiseptic were reduced by 2 fold (for *C. parapsilosis* from 0.98 to 0.48 µg/mL) or 4–8 fold (MICs alone were 0.98–7.81 µg/mL and in combination 0.48–1.95 µg/mL) for the remaining strains. 

In the case of a combination CEO and EUG with silver nitrate, a 2–4-fold reduction in the MIC value of these compounds and synergism for *C. parapsilosis* (FICI = 0.5 and 0.375 for CEO and EUG, respectively) or addition for other *Candida* spp. (FICI = 0.75) was observed (MIC = 250–500 µg/mL). The MIC of silver nitrate was also lowered by 2–8 fold in the combination compared to its MIC alone (from 7.81 to 0.98–3.91 µg/mL). 

Combining oil and its main constituent with triclosan was also a good idea. The MICs values were reduced by 2–8 fold in the combination compared to their MICs alone (MICs decreased even to 125 µg/mL for some strains). A beneficial interaction – synergism was shown for both substances in the case of *C. glabrata* and *C. parapsilosis* (FICI = 0.5). Moreover, addition with FICI = 0.625–1 was indicated for other strains. MICs of triclosan alone were 7.81–15.62 µg/mL and in combination: 3.91–7.81 µg/mL.

The indifference was shown only between these substances and nystatin, with FICI values ranging from 1.25 to 1.5. The MICs of CEO and EUG alone and its MICs in combination differed by 2–4 fold (MIC = 250–500 µg/mL and FIC = 0.25–0.5), while MICs of nystatin were the same (MIC = 0.48 µg/mL and FIC = 1). 

These results indicated an excellent effect of the combination of both natural products with all the antifungals. However, when they were combined with nystatin, no interaction was found. 

## 3. Discussion

### 3.1. The Antifungal Activity Assessment of CEO and EUG

The incidence of fungal infections has been steadily increasing in recent years [22]. *C. albicans* is a common opportunistic fungal pathogen that may cause infections, mainly in immunocompromised patients [23]. Oropharyngeal candidiasis is the most common fungal infection in these persons [24]. Moreover, the list of antimycotics or oral antiseptics is very limited. Additionally, the treatment of infections caused by *Candida* spp. is often ineffective due to their increasing resistance to antifungal agents and ability to form biofilms that protect the microorganisms from host immune defenses [25]. This situation clearly highlights the need for researching new alternative therapies. Natural products are the most promising candidates for antimycotics because they have low toxicity, low environmental impact, and a broad spectrum of action when compared to synthetic antimicrobial substances [1,2,23,26,27].

The obtained results showed the potential antifungal effect of CEO and its main component, EUG, against both reference *Candida* spp. strains (5 strains) and clinical *Candida* spp. isolates (60 isolates) from the oral cavity of hospitalized patients with hematological malignancies, which are particularly vulnerable to these infections. Our data indicated the similar antifungal activity of both compounds. The studied strains were susceptible to these phytoconstituents at MICs in the range of 0.25–2 mg/mL. The values of MFC were the same or 2–4-fold higher than MIC. Moreover, MIC_50_ = 0.5 mg/mL, MIC_90_ = 1 or 2 mg/mL, MFC_50_ = 1 mg/mL and MFC_90_ = 2 mg/mL were calculated. The largest number of *Candida* isolates were inhibited by natural compounds at a minimal concentration of 0.5 mg/mL. Additionally, their favorable fungicidal effect was observed. 

The other reports showed that CEO and EUG had similar activity to our results. According to Gucwa et al. [3], the MIC of CEO was 1.25 and 0.06 mg/mL against *C. albicans* ATCC 10231 and *C. glabrata* DSM 11226, respectively. In turn, the effect towards clinical isolates was different (MIC = MFC = 0.1–26.25 mg/mL) with the highest frequency MIC and MFC in the range of 0.42 to 13.12 mg/mL for *C. albicans*. In the case of *C. glabrata*, these minimal concentrations were similar (MIC = MFC = 0.21–26.25 mg/mL). The values of MIC = 0.42–13.12 mg/mL and MFC = 1.68–6.56 mg/mL were the most common. Similar results were indicated by Rajkowska et al. [28] for CEO with MIC = 0.5 mg/mL towards *C. albicans*. In the data of Satthanakul et al. [25], MIC = 0.25–0.5 and MFC = 1 mg/mL against *C. albicans, C. krusei* and *C. tropicalis* for CEO were found. In turn, Khan et al. [29] exhibited much better MIC values of this EO in the range of 0.05–0.4 mg/mL against *C. albicans* isolates.

In the case of EUG, the results were similar. Sharifzadeh and Shokri [17] showed the antifungal potential of EUG against *C. tropicalis* strains (MIC = 0.4–0.8 mg/mL) and *C. krusei* (MIC = 0.2–0.4 mg/mL). According to Ahmad et al. [18], MICs of EUG against reference and clinical *Candida* spp. strains range from 0.475 to 0.5 mg/mL. The research carried out by Silva et al. [30] showed that EUG inhibited the cellular growth of *C. albicans* (ATCC 90028) at MIC = 0.625 mg/mL. The subsequent studies [31] indicated a slightly higher activity of *o-*eugenol with MIC = 64–128 and 32–128 µg/mL against *C. albicans* and *C. tropicalis*, respectively. All data [17,18,30,31] confirmed the anticandidal effect of both phytoconstituents. 

### 3.2. Mode of Antifungal Action of CEO and EUG

The mechanism of antimicrobial action of EOs is complex and depends on their chemical composition and the quantity of the major single compounds. Some reports revealed that constituents of EOs mixture could cause cell membrane damage, influence many other cellular activities, including energy production, may be linked to reduced membrane potentials, the disruption of proton pumps, and the depletion of the ATP, the coagulation of cell content, cytoplasm leakage, and finally cell apoptosis or necrosis, leading to cell death (Figure 5) [3]. 

Therefore, the next step of our research was to evaluate the mechanism of action of the studied oil and its main compound on the cell of yeast from *Candida* spp. The aim of these studies was to determine whether their antifungal activity involves a direct interaction with the cell wall structure and/or with the ion permeability of the membrane of these microorganisms [28,32,33,34,35].

In general, it would appear that essential oils act on several levels, depending on the concentration of the oil. The plasma membrane and the cell wall appear to be particularly affected. Various studies also show a loss of membrane integrity, a decrease in the amount of ergosterol (major component of the fungal membrane), and an inhibition in wall formation. Essential oils also have an inhibitory action on membrane ATPases and cytokine interactions, the mitochondria and the endoplasmic reticulum appear to be important sites in their mechanisms of action. Finally, the expression of a certain number of genes seems to be affected, notably genes involved in adhesion, growth, dimorphism, sporulation, etc. Only preliminary data are currently known to explain the mechanisms of action of other compounds [19].

First, the effects of the CEO and EUG on the cell wall of fungi was tested. The fungal cell wall is a dynamic structure that protects the cell from changes in osmotic pressure and other environmental stresses while allowing the fungal cell to interact with its environment. The structure and biosynthesis of a fungal cell wall is unique to the fungi, and is therefore an excellent target for the development of antifungal drugs. In turn, sorbitol is a factor that causes slight cell stress, which may cause the inhibition of cell growth in the presence of some nonspecific cell wall inhibitors [3]. It is used to stabilize fungi protoplasts, protecting their cell wall from environmental stresses, particularly osmotic changes [28,32,34]. It is expected that the MIC of a compound that damages the cell wall will shift to a much higher value in the presence of the osmotic support. This assay showed that the MIC of the studied oil and its component did not vary in the presence of sorbitol for any of the yeasts tested, suggesting that both natural compounds would not act by inhibiting the mechanisms controlling cell wall synthesis, similarly to nystatin. Both CEO, EUG and nystatin, at evaluated concentrations, inhibited the cellular growth of *Candida* spp. strains and the presence of an osmotic protector did not interfere with their antifungal effect. Similar results were obtained by Silva et al. [30] for EUG and nystatin against *C. albicans* ATCC 900280. According to them, the MIC of EUG was 0.625 mg/mL in the presence and absence of sorbitol. According to Rajkowska et al. [29], the MICs of CEO were also independent of the sorbitol level in the medium (0.5 mg/mL). These values stayed unchanged what suggests lack of their influence on cell wall structure. On the other hand, Gucwa et al. [3] showed quite different and inconsistent results—in the presence of 0.8 M sorbitol, the MIC value of CEO for *C. albicans* ATCC 10231 was significantly reduced, by 4 fold (from 1.25 to 0.31 mg/mL); and in the case of *C. glabrata* DSM 11226, the MIC value increased by 4 fold (from 0.6 to 2.5 mg/mL). 

Subsequent studies included determining whether CEO and EUG act by affecting ergosterol in the fungal cell membrane. Ergosterol (the major sterol component of the fungal cell membrane) and enzymes of the ergosterol biosynthetic pathway are very important targets of several classes of antifungals such as polyenes and azoles [34]. These drugs are especially often used for the treatment of candidiasis. Ergosterol is a fungal-specific structure responsible for maintaining cell function and integrity [36]. It plays an essential role in many cellular processes of fungi, including regulation of membrane proteins, endocytosis, cell division, membrane fluidity and cell signaling. The basis for this research is that an exogenous source of ergosterol in the medium may increase the MIC value for compounds that target this sterol in the cell membrane [3]. The obtained data showed a 2–8-fold increase in the MIC of both CEO and EUG in the medium with exogenous sterol compared with the sterol-free experiment against all reference yeasts. The cell membrane of *Candida* spp. strains may be more or less susceptible to the action of these compounds. In the case of the CEO, an 8- and 2-fold increase was observed, respectively, for *C. albicans* ATCC 10231 and *C. glabrata* ATCC 90030. In turn, the 8-fold increase in MIC for EUG towards *C. parapsilosis* ATCC 22019 also indicates its significant role in cell membrane disintegration. Similarly, an 8–16-fold increase in MIC was observed for polyene antibiotic–nystatin, which binds the ergosterol found in lipid bilayer membranes and whose interaction with fungal cell membrane ergosterol is already known [2,3,28,32,35]. The most sensitive to nystatin (MIC of 16-fold higher) was the cell membrane of both reference *C. albicans* strains. These data are consistent with previous studies of Castro and Lima [34], in which the MIC value of this antibiotic against *C. albicans* also increased by 16-fold in the presence of sterol. These data suggest that studied natural compounds appear to bind to the ergosterol in the membrane, which increases the permeability of Ca^2+^, and K^+^ ions, radicals, and proteins [2,33,34] and thus destabilization of the cell membrane, inhibition of fungal growth and ultimately results in cell death. Cell membrane integrity is essential for the survival of fungi as it is responsible for cell function [31]. Ergosterol assay data suggest that CEO and EUG may act at the level of fungal cell membrane. Perhaps it is related to their antifungal activity. According to Pinto et al. [19], CEO and EUG considerably impair the biosynthesis of ergosterol in *C. albicans*, *C. glabrata* and *C. tropicalis* strains. The data of Gucwa et al. [3] showed similar results. In the ergosterol-containing medium, the MIC values of CEO for *C. albicans* ATCC 10231 and *C. glabrata* DSM 11226 increased from 1.25 mg/mL and 0.6 to ≥ 2.5 mg/mL (at least 2 to 4-fold), respectively. In their studies, amphotericin B (also known to act on membrane ergosterol) was used as a positive control. For *C. albicans* ATCC 10231, the MIC of this polyene increased from 0.06 to 0.5 mg/mL (8-fold increase), and for *C. glabrata* DSM 11226 – from 0.06 to 8 mg/mL (increased 128-fold) [3]. In the ergosterol binding assay, according to Rajkowska et al. [28], the MIC of CEO in the presence of ergosterol was 8-fold higher than the corresponding MIC without ergosterol (from 0.5 to 8 mg/mL). Their results also suggested that this oil may inhibit yeast growth through binding to sterol. These authors reported additionally similar mechanism of action for tea tree, thyme and peppermint oils. Further research by Gucwa et al. [3] revealed that thyme, lemon and geranium oils also influence the cell membranes. Up to a 16-fold increase in MIC value was observed for lemon oil for the *C. glabrata* strain. In the case of *C. albicans*, the change was not so noticeable and reached a 2-fold increase. The efficacy in a reduction in the total cellular ergosterol content in *C. albicans* was also observed for cinnamaldehyde, furfuraldehyde, citral, indole, piperide, α-pinene and β-pinene [37].

### 3.3. Investigation of Interaction of CEO and EUG with Selected Antimycotics

It is well known that *Candida* species can colonize both hard and soft tissues. Therefore, with regard to oral environment, *Candida* can participate not only in the pathogenesis of oral candidiasis but also in endodontic, periodontal and peri-implant infections. Clinically, it is suggested that CEO and/or EUG could be used as an adjuvant substance to control these infections, especially in endodontic infections [30]. Moreover, with regard to oral candidiasis, a local antimicrobial agent is preferred, in order to produce better efficacy due to higher drug penetration and retention, also avoiding systemic side effects. Regarding the clinical use of EUG, it is considered a therapeutic agent that is particularly widely used in dentistry and it is assumed that its combination with some antifungals used in oral candidiasis may improve the effectiveness of both [30]. Therefore, the next stage of this research was to evaluate the interaction between CEO and EUG with selected antimycotics or antiseptics in order to determine the potential synergistic effect with them. Antimicrobial combination therapy involves simultaneous use of two or more compounds, often with a different mode of action, which may lead to synergy. A synergistic effect implies that a combination of these substances shows substantially higher activity than each compound individually [38]. Synergy can be achieved often if the mixed compounds are able to affect different target sites in cells of microorganism, or they may interact with one another to increase solubility, thereby enhancing bioavailability. The combined formulation also has the potential to decrease toxicity and adverse side effects by lowering the required dose of individual components [39]. Most EOs are used in blends or combinations of two or more oils or with other antimicrobial substances [39].

According to literature data [25,39,40,41], CEO and/or EUG were mixed with some antimicrobial compounds to obtain a synergistic antifungal effect. In our studies, selected synthetic antiseptics—cetylpyridinium chloride monohydrate, chlorhexidine, silver nitrate or triclosan and polyene antibiotic—nystatin were used to assess interaction with CEO and EUG towards reference *Candida* spp.: *C. albicans, C. glabrata, C. krusei* and *C. parapsilosis*. In the field of dentistry, these compounds are used in professional oral hygiene, and prevention or treatment of oral infections, they are also included in lots of oral care products such as mouthwashes or dentifrices. They are also used to inhibit fungi and biofilm formation in dental products or are promising strategy for caries control, particularly for young children, the elderly or patients with severe caries risk [25,38,39,40,41].

The obtained results, carried out in vitro using the checkerboard method, indicated a very good effect of their combination with all antiseptics used in studies. The tested natural compounds indicates both synergistic and additive interactions with CEO and EUG. In the case of nystatin, no interaction was found. Moreover, no antagonistic effect towards *Candida* strains was observed. The MIC values of CEO and EUG alone were 1000 µg/mL for all strains. In turn, in the combination, their MICs were reduced by 2–16-fold depending on the compound and *Candida* strain. 

Both compounds showed synergy with cetylpyridinium chloride against all *Candida* spp. strains with FICI = 0.375–0.5. Their MICs and MICs of antiseptics in combination decreased by 4–8 fold. In the literature data, there are no studies evaluating the interaction of EOs with this antimicrobial agent. However, the mode of action of cetylpyridinium chloride may contribute to the increase in the activity of EUG. This compound affects the cell by interfering with its osmoregulation and its homeostasis, measurably proven by K^+^ and pentose leakage in yeasts, which might initiate autolysis by activation of intracellular latent ribonucleases. Moreover, it leads to disintegration of the membranes with subsequent leakage of cytoplasmic contents. Damage to proteins and nucleic acids, as well as cell wall lysis by autolytic enzymes are the consequences [41].

The very favorable post-combination effect was also shown for CEO and EUG with chlorhexidine (synergism at FICI = 0.375–0.5) for all strains except *C*. *parapsilosis* (addition at FICI = 0.562). Strains were even 4–16-fold more sensitive to EO and its component in combination. In turn, MICs of chlorhexidine were reduced by 2–8 fold for the individual strains. Similar studies were performed by Satthanakul et al. [25]. These authors observed that the combination of CEO and chlorhexidine reduced the MICs of both the EO and antiseptic against *C. albicans, C. krusei* and *C. tropicalis*. Synergistic effect (FICI ≤ 0.5) was found for combinations of EO with chlorhexidine against *C. albicans* and *C. tropicalis* and addition against *C. krusei*. Their synergistic activity might be due to target site. Chlorhexidine is believed to act by binding to proteins in the cell wall leading to a loss of cell integrity, the leakage of cell constituents and cell precipitation. Thus, the hydrophobic properties of EO might enable them to penetrate the lipid bilayer of the cell membrane and alter the membrane structure, which may enhance cell permeability to chlorhexidine [40].

Synergistic effects against sessile *C. albicans* in biofilm were also found for CEO combined with chlorhexidine (FICI = 0.5). These combinations reduced the SMIC_50_ (sessile MIC_50_) of both CEO and antiseptic by 4-fold. In the case of *C. krusei* and *C. tropicalis*, there was an additivity effect with FICI = 0.75 and 1.25, respectively. The synergism of CEO in this combination against sessile *C. albicans* in biofilm may be due to the penetration of the hydrophobic EO components through the charged extracellular matrix of the biofilm. If the cytoplasmic membrane of the sessile *Candida* cells was disturbed, this might increase uptake of chlorhexidine (and also the EO components) into the target sites at cell membrane causing cell membrane damage, loss of structural organization and integrity and coagulation of cytoplasmic constituents [40].

These data suggested, like our results, that the combination of CEO with chlorhexidine was more effective against *Candida* spp. than CEO alone. This mix can lower the amount of CEO and chlorhexidine required to treat an infection and reduce their side effects (in the case of chlorhexidine, tooth staining, bitter taste and burning sensation) [40]. 

In the case of combination CEO and EUG with silver nitrate, a 2–4-fold reduction in the MIC value and synergism for *C. parapsilosis* (FICI = 0.5 and 0.375 for CEO and EUG, respectively) or addition for other *Candida* spp. (FICI = 0.75) was observed. The MICs of silver nitrate were also lowered by 2–8 fold in the combination. The results of Thilakan et al. [41] showed that silver nitrate is highly effective against *C. albicans*, probably through the destruction of membrane integrity. However, a broad understanding of the mechanisms underlying the antifungal activity of silver nitrate is still to be identified. There have also been no studies evaluating the interaction of EO with silver nitrate. 

A synergistic effect was also demonstrated for triclosan, whose MICs decreased by 2–4-fold in the combination compared to its MICs alone. In turn, MICs of natural compounds were 2–8-fold lower (FICI = 0.5 for *C. glabrata* and *C. parapsilosis*). The addition with FICI = 0.625–1 was indicated for remaining strains. It was showed by Alfhili et al. [42] that triclosan leads to K^+^ leakage indicating membrane damage, inhibits membrane-bound ATPase enzymatic activity, causes membrane destabilization, perturbs ion transport, and modulates the overall osmoregulation of cell [42]. This may contribute to the enhanced antifungal effect of CEO and EUG in combination with triclosan. 

The indifference was showed only between these substances and nystatin with FICI values ranging from 1.25 to 1.5. The MIC of CEO and EUG alone and its MICs in combination differed by 2–4-fold, while MICs of nystatin were the same (FIC = 1). Similar results were obtained by Silva et al. [30]. These authors investigated antifungal interaction of EUG and nystatin towards *C. albicans*. MICs of EUG alone and with nystatin were the same and persisted at 625 μg/mL. In turn, the MIC of nystatin decreased from 25 μg/mL to 3.125 μg/mL in the presence of EUG. Antifungal interactions between these compounds were determined by FICI calculation (FICI = 1.125) and considered indifferent. 

Some authors [17,43,44,45] tested also interaction between EUG and other antifungal drugs. The combined effect of EUG and fluconazole or amphotericin B on single and mixed biofilm cells was determined by Jafri et al. [43]. The interaction between EUG and these antifungals was studied against the reference *C. albicans*. Synergy was observed between EUG and fluconazole with FICI values ranging from 0.156 to 0.25. However, indifferent interaction was noticed for EUG and amphotericin B (FICI = 0.625) [43].

In turn, Sharifzadeh and Shokri [17] reported the antifungal potential of EUG and its interaction with voriconazole against *Candida* strains. MIC values of EUG were 400–800 and 200–400 µg/mL against *C. tropicalis* and *C. krusei*, respectively. They observed and confirmed synergistic effects of EUG with voriconazole towards these yeasts.

The subsequent research was carried out by Ahmad et al. [18]. According to them, the MICs of EUG against various clinical fluconazole-sensitive and -resistant *Candida* strains including five American Type Culture Collection (ATCC)-type strains (*C. albicans* ATCC 10261, *C. albicans* ATCC 90028, *C. albicans* ATCC 44829, *C. tropicalis* ATCC 750 and *C. glabrata* ATCC 90030) range from 475 to 500 µg/mL. FICI values for EUG with fluconazole combinations against all fluconazole-sensitive *Candida* isolates ranged from 0.31 to 0.55 and for fluconazole-resistant *Candida* isolates from 0.39 to 0.68. The interaction between EUG and this drug showed a high amount of synergism. Moreover, other authors [45] showed for EUG in vitro synergy with fluconazole and amphotericin B against *C. albicans.*


Other authors [39] indicated synergy of CEO in combination with rosemary oil (*Rosmarinus officinalis* L.) and addition (FICI = 0.58) of lavender EO (*Lavandula angustifolia* L.) towards *C. albicans* (ATCC 10231). 

The findings reported in this paper indicate the high antifungal potential of CEO and EUG against both the reference and clinical *Candida* spp. strains. These result are very interesting from the point of view of their potential use as an alternative for conventional prevention and treatment, especially superficial candidiasis. Additionally, experiments confirming the possibility of a synergistic or additive effect with antiseptics allow us to conclude that these natural substances can be used as a supplement for oral hygiene, prevention and treatment of oral candidiasis.

## 4. Materials and Methods

### 4.1. Materials

#### 4.1.1. The Studied Compounds

In these studies, clove essential oil (CEO) (Dr Beta, Poland) and its main constituent, eugenol (EUG) (Sigma-Aldrich Chemicals, St. Louis, Mo., USA), were used to assess the antifungal activity against reference and clinical isolates of *Candida* spp. Additionally, selected antimycotics (antiseptics or antibiotic)—silver nitrate, cetylpyridinium chloride monohydrate, triclosan (5-chloro-2-(2,4-dichlorophenoxy)phenol; 2,4,4′-trichloro-2′-hydroxydiphenyl ether) (Glentham Life Sciences, Great Britain), chlorhexidine and nystatin (Sigma-Aldrich Chemicals, St. Louis, Mo., USA)—were used for evaluation of the interactions with CEO and EUG. All compounds were dissolved in dimethyl sulfoxide (DMSO) (Pol-Aura, Różnowo, Poland) to obtain a stock solution for use in vitro tests. 

#### 4.1.2. Microorganisms

The reference strains of fungi from American Type Culture Collection (ATCC) were included. These fungi belonged to yeasts: *Candida albicans* ATCC 2091, *Candida albicans* ATCC 10231, *Candida parapsilosis* ATCC 22019, *Candida glabrata* ATCC 90030 and *Candida krusei* ATCC 14243. Moreover, 60 clinical isolates of *Candida* spp.: *C. albicans* (30 isolates) and non-*albicans Candida* spp. (NAC) (30 isolates: *C. tropicalis, C. glabrata, C. famata, C. parapsilosis, C. krusei, C. guilliermondii,* and *C. lusitaniae*) were used. These microorganisms were isolated from the oral cavity of hospitalized patients with hematological malignancies (from the collection of clinical strains deposited in the Department of Pharmaceutical Microbiology of Medical University in Lublin, Poland). The Ethical Committee of the Medical University of Lublin approved the study protocol (No. KE-0254/75/2011). The isolates were identified by standard diagnostic methods—microscopic, macroscopic, and biochemical microtest, e.g., ID 32 C (bioMèrieux S.A., Warsaw, Poland)—on the basis of assimilation of various substrates [37]. Strains were stored as glycerol stock at −70 °C. For research purposes, fungal cultures were conducted at 35 °C for 24 h on Sabouraud agar (BioMaxima S.A., Lublin, Poland).

### 4.2. Methods

#### 4.2.1. Attenuated Total Reflection–Fourier Transform Infrared (ATR–FTIR) 

ATR–FTIR spectra of clove essential oil and eugenol were recorded on a Bruker Tensor 27 FTIR spectrometer (Bruker Optics GmbH & Co. KG, Ettlingen, Germany) equipped with single-bounce diamond ATR (Platinum ATR, Bruker Optics GmbH & Co. KG, Ettlingen, Germany). The spectrometer was controlled with the software OPUS 6.5 (Bruker Optics GmbH & Co. KG, Ettlingen, Germany). The scan number of the spectra was 16, recorded at a 4 cm^−1^ resolution in the wavenumber range from 4000 to 400 cm^−1^. A small amount of each samples (approximately 10 mL) was placed on the ATR surface that was cleaned using ethanol to eliminate any contamination by the previous sample. A new background was recorded between each replicate, and the scans were run in triplicates [20].

#### 4.2.2. Gas Chromatography

Qualitative and quantitative analysis of the essential oil was performed in accordance with the previously described methodology [46] using a Shimadzu GC-2010 Plus instrument coupled to a Shimadzu QP2010 Ultra mass spectrometer (Shim-Pol, Izabelin, Poland). Compounds were separated on a fused-silica capillary column ZB-5 MS (30 m, 0.25 mm i.d.) with a film thickness of 0.25 mm (Phenomenex, Torrance, CA, USA). The following oven temperature program was initiated at 50 °C, held for 3 min, then increased at the rate of 8–250 °C/min, and held for a further 2 min. The spectrometers were operated in the EI mode; the scan range was 40–500 amu, the ionization energy was 70 eV, and the scan rate was 0.20 s per scan. The injector, interface, and ion source were kept at 250, 250, and 220 ℃, respectively. Split injection was conducted with a split ratio of 1:20, and helium was used as the carrier gas at a 1.0 mL/min flow rate. EO samples were prepared by diluting 2 µL in 1 mL of hexane. The relative percentages of each component present in the analyzed EO were calculated. The retention indices were determined in relation to a homologous series of n-alkanes (C8–C24) under the same operating conditions. Compounds were identified using computer-assisted spectral libraries (NIST 2011, Gaithersburg, MD, USA).

#### 4.2.3. *In Vitro* Antifungal Activity Assay of CEO and EUG

To verify the antifungal activity of the natural compounds CEO and EUG against reference and clinical strains of *Candida* spp., the broth microdilution method was used according to European Committee on Antimicrobial Susceptibility Testing (EUCAST) [47] and Clinical and Laboratory Standards Institute (CLSI) guidelines [48]. These assays were performed as described previously [36]. The compounds CEO and EUG were dissolved in DMSO to obtain a stock solution with a concentration of 50 mg/mL. The MIC (minimal inhibitory concentration) of the compounds was examined using their 2-fold dilutions in RPMI (Roswell Park Memorial Institute) 1640 broth (Sigma-Aldrich Chemicals, St. Louis, Mo., USA) with MOPS (3-(N-Morpholino)propanesulfonic acid) (Sigma-Aldrich Chemicals, St. Louis, Mo., USA), prepared in 96-well polystyrene plates. Final concentrations of CEO and EUG ranged from 8 to 0.625 mg/mL. All of the used strains of yeasts were first subcultured on Sabouraud agar at 37 °C for 24 h. The fungal suspensions were prepared in sterile saline (0.85% NaCl) with an optical density of McFarland standard scale 0.5—approximately 5 × 10^5^ CFU/mL (Colony Forming Units/mL). Next, to each well containing 100 µL of RPMI 1640 broth with MOPS and the above various concentrations of tested compounds, 1 µL of the appropriate fungal suspension was added. After incubation (37 °C, 24 h), their MIC was assessed spectrophotometrically as the lowest concentration of the samples showing complete fungal growth inhibition. The DMSO, growth and sterile controls were also carried out. The standard antifungal antibiotic–nystatin (Sigma-Aldrich, Chemicals, St. Louis, Mo., USA) was used as positive control. In turn, the MFC (minimal fungicidal concentration), defined as the lowest concentration of the compounds, was required to kill a particular fungal species. MFC was determined by removing the culture using for MIC determinations from each well and spotting onto Sabouraud agar medium. Then, the plates were incubated in the appropriate conditions (as before). The lowest compounds concentrations with no visible growth observed was assessed as a fungicidal concentration. All the experiments were repeated three times and representative data are presented. Moreover, the MFC/MIC ratios were also calculated in order to determine the fungicidal (MFC/MIC ≤ 4) or fungistatic (MFC/MIC > 4) effect of the studied natural compounds [49,50].

#### 4.2.4. Mode of Antifungal Action of CEO and EUG

##### Sorbitol Assay

To evaluate the effect of CEO and EUG on the cell wall of *C. albicans*, the sorbitol assay was performed in accordance with the procedure reported by other authors [2,28,32,33,34]. The sorbitol (Sigma-Aldrich Chemicals, St. Louis, Mo., USA) was added to the culture medium in a final concentration of 0.8 M. The MIC of the tested natural compounds using Sabouraud Dextrose Broth (SDB) medium (BioMaxima S.A., Lublin, Poland) with and without sorbitol against yeasts was determined. The test was performed using the microdilution technique in triplicate according to the previous guidelines [49,50]. After filling each well of the microplates with 100 µL of SDB with and without sorbitol, serial dilutions of studied compounds and nystatin (as control) ranging from 8 to 0.0625 mg/mL and from 1000 to 0.004 μg/mL were carried out, respectively. Subsequently, 10 µL of yeast suspension was added to each well. Yeast growth and sterility control were also performed. The plates were incubated at 37 °C and MIC was read after 2 and 7 days [2,28,32,33,34].

##### Ergosterol Assay

To assess if CEO and EUG bind to the fungal membrane sterols, this experiment was performed according to the procedure described by other authors [2,28,32,33,34]. First, the stock solution of exogenous ergosterol (Sigma-Aldrich Chemicals, St. Louis, Mo., USA) at final concentration 10 mg/mL was prepared. The MIC of tested compounds against *C. albicans* was determined by using broth microdilution techniques according to the previous guidelines [49,50] in the presence and absence of exogenous ergosterol (in a final concentration of 400 μg/mL), added to the assay medium. After filling each well of the microplates with 100 µL of SDS medium with and without ergosterol, serial dilutions of the examined natural substances and nystatin (as control) ranging from 8 to 0.0625 mg/mL and from 1000 to 0.004 μg/mL were carried out, respectively. Then, 10 µL of yeast suspension was added to each well. The plates were incubated at 37 °C for 24 h and MIC was assessed. Yeast growth and sterility were also controlled [2,28,32,33,34,42].

#### 4.2.5. Investigation of Interaction of the Natural Compounds and Selected Antimycotics

To determine the fractional inhibitory concentrations (FICs) of CEO and EUG in combination with other antifungal substances a checkerboard technique (according to CLSI) was used in 96 well microtiter plates. Different antimycotics for these studies were used: antibiotic—nystatin; synthetic antiseptics—chlorhexidine; silver nitrate; cetylpyridinium chloride monohydrate; triclosan; and chlorhexidine (Sigma-Aldrich Chemicals, St. Louis, Mo., USA). The tested natural substances and selected antimycotics were used at specific concentrations (estimated at their respective MIC values). These compounds at various concentrations in the broth corresponding to 8 × MIC, 4 × MIC, 2 × MIC, MIC, 1/2 × MIC, 1/4 × MIC, and 1/8 × MIC (from 8-fold greater than their MIC to 8-fold lower than their MIC) were added horizontally (CEO and EUG) and vertically (selected antifungals) to the wells of the plate. Finally, the reference *C. albicans, C. glabrata, C. krusei* and *C. parapsilosis* inoculums were added to per each well in the plate. Growth and sterility controls were also performed. Next, the plates were incubated at 37 °C for 24 h. Each test was performed in triplicate [2,21,34]. After describing the MIC for each row, the FIC and Ʃ FIC (FIC index) were calculated as: Ʃ FIC = FICA + FICB = (CA/MICA) + (CB/MICB), where MICA and MICB are the MICS of compounds A (natural compounds) and B (selected antimycotics) alone, respectively. In turn, CA and CB are the concentrations of the studied compounds: A in combinations with B and B in combinations with A, respectively. FICI–FIC index values were interpreted as follows: FICI values of ≤ 0.5 as synergy, FICI values between 0.5 and 1 as additive, FICI values between 1 and 4 as indifferent, and FICI values > 4 as antagonism [2,21,34].

#### 4.2.6. Data Analysis

All the samples were analyzed in triplicate and representative data (mode) are presented. 

## 5. Conclusions

These results showed satisfactory susceptibility in vitro of both reference and clinical *Candida* spp. strains from the oral cavity of hospitalized patients with hematological malignancies to the studied CEO and its main compound, EUG. These natural plant compounds appear to bind to the ergosterol in the membrane, which increases ion permeability and ultimately results in cell death. Additionally, a stronger antifungal action of CEO and EUG can be obtained by combining it with all the antifungals, especially with chlorhexidine and cetylpyridinium chloride, due to the favorable synergistic interactions. It may increase their effectiveness and find wide application in the treatment of surface candidiasis. On the basis of the obtained data, it is concluded that these compounds alone or in combination with other antimycotics can be used as components of antifungal preparations used topically in the prevention and treatment of superficial, especially oral candidiasis. However, this requires further in vivo studies.

## Figures and Tables

**Figure 1 molecules-28-00215-f001:**
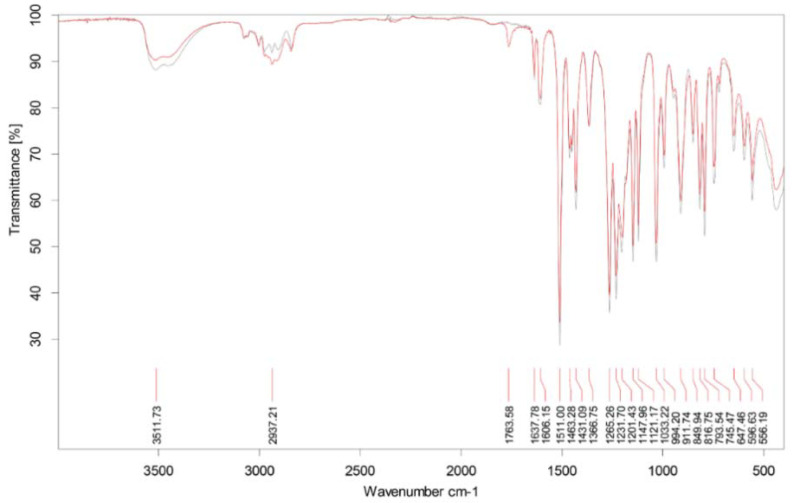
ATR-FTIR spectra for CEO and EUG.

**Figure 2 molecules-28-00215-f002:**
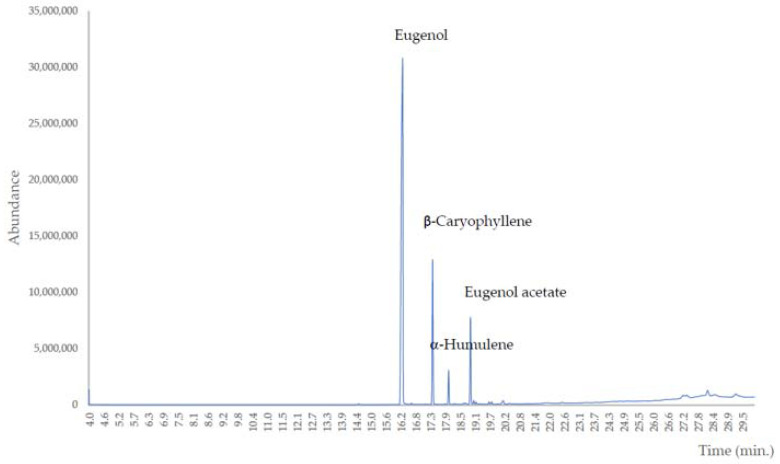
GC–MS chromatogram of CEO.

**Figure 3 molecules-28-00215-f003:**
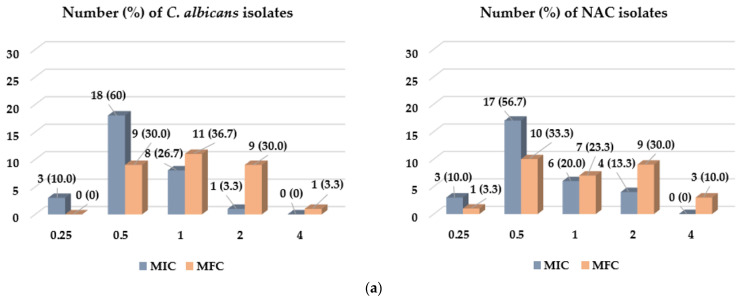
Distribution of MIC (minimal inhibitory concentration) and MFC (minimal fungicidal concentration) values [mg/mL] of CEO (**a**) and EUG (**b**) among clinical isolates of *C. albicans* (30 isolates) and non-*albicans Candida* spp. (NAC) (30 isolates) from hospitalized patients with hematological malignancies.

**Figure 4 molecules-28-00215-f004:**
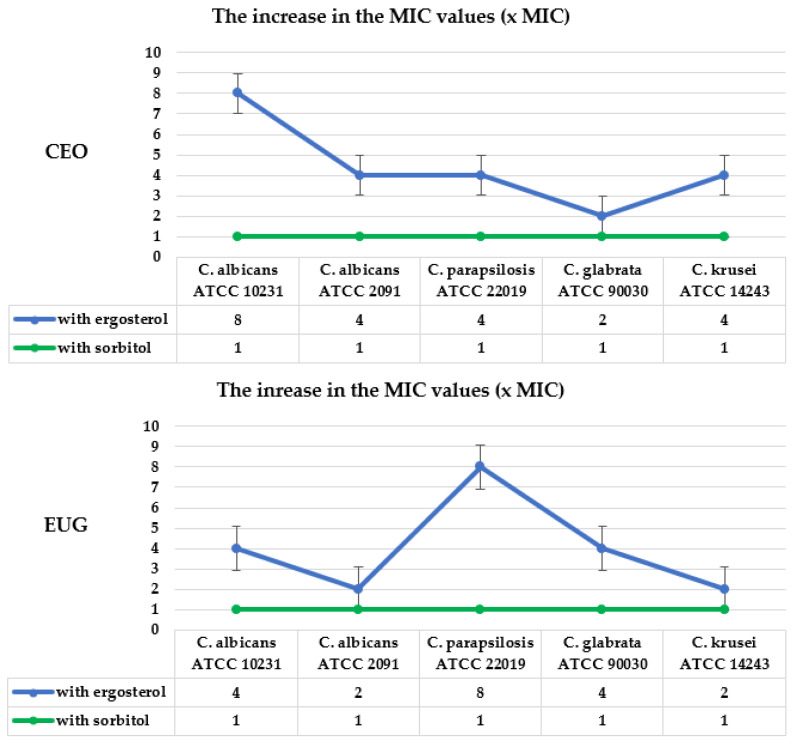
The increase in the MIC values (x MIC) of CEO, EUG and NYS (as control) in the presence of ergosterol (400 µg/mL) and sorbitol (0.8 M) against reference *Candida* spp.

**Figure 5 molecules-28-00215-f005:**
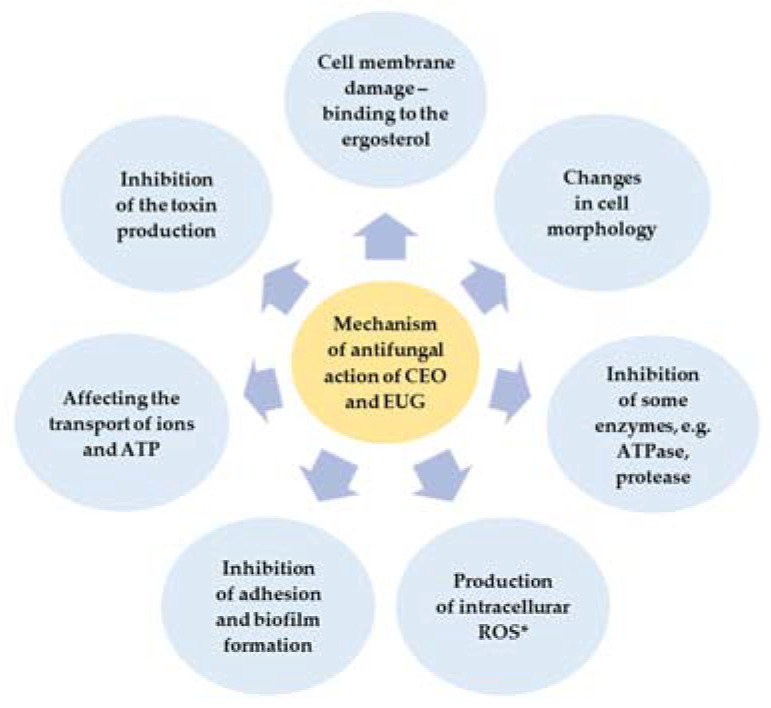
The mechanism of antifungal action of CEO and EUG (ROS*—Reactive Oxygen Species) (own elaboration based on [2,3,19,28,32,33,34,35,36]).

**Table 1 molecules-28-00215-t001:** Relative percentages of the main volatile components identified in CEO.

No	Retention Time (min.)	Retention Index *	Composition (%)	Compound
1	16.250	1366	66.81	Eugenol
2	17.423	1433	14.11	β-Caryophyllene
3	18.047	1469	3.19	α-Humulene
4	18.903	1521	8.33	Eugenol acetate

* Retention indices relative to C8-C24 n-alkanes on the ZB-5 MS column.

**Table 2 molecules-28-00215-t002:** The activity data of CEO and EUG expressed as a range of MIC (minimal inhibitory concentration) or MFC (minimal fungicidal concentration) [mg/mL] and MFC/MIC ratio against the reference strains of *Candida* spp.

Reference Strains	CEO			EUG		
	Range of MIC	Range of MFC	MFC/MIC Ratio	Range of MIC	Range of MFC	MFC/MIC Ratio
*C. albicans* ATCC 10231	1–2	2–4	1–2	1–2	2–4	2
*C. albicans* ATCC 2091	0.5–1	1–2	2	0.5–1	1–2	2
*C. parapsilosis* ATCC 22019	1–2	2–4	2	0.5–1	1–2	2
*C. glabrata* ATCC 90030	1–2	2–4	2	1–2	2–4	2
*C. krusei* ATCC 14243	0.5–2	1–4	2	0.25–1	1–2	2

**Table 3 molecules-28-00215-t003:** The activity data of CEO and EUG expressed as a range of MIC (minimal inhibitory concentration) or MFC (minimal fungicidal concentration) [mg/mL] and MIC_50_/MIC_90_ ratio or MFC_50_/MFC_90_ ratio against clinical isolates of *C. albicans* (30 isolates) and non-*albicans Candida* spp. (NAC) (30 isolates) from hospitalized patients with hematological malignancies.

Clinical Isolates	CEO				EUG			
	Range of MIC	Range of MFC	MIC_50_ /MIC_90_	MFC_50_ /MFC_90_	Range of MIC	Range of MFC	MIC_50_ /MIC_90_	MFC_50_ /MFC_90_
*C. albicans*	0.25–2	0.5–4	0.5/1	1/2	0.25–2	0.5–4	0.5/1	1/2
non*-albicans Candida* spp.	0.25–2	0.25–4	0.5/2	1/2	0.25–2	0.5–4	0.5/1	1/2

**Table 4 molecules-28-00215-t004:** The fungicidal effect of CEO and EUG expressed as a MFC/MIC (minimal fungicidal concentration/minimal inhibitory concentration) ratio against clinical isolates of *C. albicans* (30 isolates) and non-*albicans Candida* spp. (30 isolates) from hospitalized patients with hematological malignancies.

MFC/MIC Ratio	Number (Percentage) of Clinical Isolates			
	*C. albicans*		non-*albicans Candida* spp.	
	CEO	EUG	CEO	EUG
1	8 (26.7)	6 (20)	11 (36.7)	9 (30)
2	19 (63.3)	20 (66.7)	15 (50.0)	18 (60)
4	3 (10.0)	4 (13.3)	4 (13.3)	3 (10)

**Table 5 molecules-28-00215-t005:** MICs and FIC indices of CEO alone and in combination with selected antimycotics on reference *C*. *albicans* ATCC 10231 strain.

Antifungal Agent	MIC of Antifungal Agent (µg/mL) against *C. albicans*		FIC	Ʃ FIC (FICI)	Interpretation
	Alone	Combination			
CEO	1000	500	0.5	1.5	indifference
nystatin	0.48	0.48	1		
CEO	1000	250	0.25	0.5	synergism
cetylpyridinium	3.91	0.98	0.25		
CEO	1000	250	0.25	0.375	synergism
chlorhexidine	7.81	0.98	0.125		
CEO	1000	500	0.5	1	addition
silver nitrate	7.81	3.91	0.5		
CEO	1000	500	0.5	1	addition
triclosan	7.81	3.91	0.5		

**Table 6 molecules-28-00215-t006:** MICs and FIC indices of CEO alone and in combination with selected antimycotics on reference *C*. *glabrata* ATCC 90030 strain.

Antifungal Agent	MIC of Antifungal Agent (µg/mL) against *C. glabrata*		FIC	Ʃ FIC (FICI)	Interpretation
	Alone	Combination			
CEO	1000	250	0.25	1.25	indifference
nystatin	0.48	0.48	1		
CEO	1000	250	0.25	0.5	synergism
cetylpyridinium	0.98	0.24	0.25		
CEO	1000	250	0.25	0.5	synergism
chlorhexidine	7.81	1.95	0.25		
CEO	1000	250	0.25	0.75	addition
silver nitrate	7.81	3.91	0.5		
CEO	1000	250	0.25	0.5	synergism
triclosan	15.62	3.91	0.25		

**Table 7 molecules-28-00215-t007:** MICs and FIC indices of CEO alone and in combination with selected antimycotics on reference *C*. *krusei* ATCC 14243 strain.

Antifungal Agent	MIC of Antifungal Agent (µg/mL) against *C. krusei*		FIC	Ʃ FIC (FICI)	Interpretation
	Alone	Combination			
CEO	1000	500	0.5	1.5	indifference
nystatin	0.98	0.98	1		
CEO	1000	250	0.25	0.5	synergism
cetylpyridinium	0.98	0.24	0.25		
CEO	1000	125	0.125	0.375	synergism
chlorhexidine	1.95	0.48	0.25		
CEO	1000	250	0.25	0.75	addition
silver nitrate	7.81	3.91	0.5		
CEO	1000	250	0.25	0.75	addition
triclosan	15.62	7.81	0.5		

**Table 8 molecules-28-00215-t008:** MICs and FIC indices of CEO alone and in combination with selected antimycotics on reference *C*. *parapsilosis* ATCC 22019 strain.

Antifungal Agent	MIC of Antifungal Agent (µg/mL) against *C. parapsilosis*		FIC	Ʃ FIC (FICI)	Interpretation
	Alone	Combination			
CEO	1000	250	0.25	1.25	indifference
nystatin	0.48	0.48	1		
CEO	1000	125	0.125	0.375	synergism
cetylpyridinium	1.95	0.48	0.25		
CEO	1000	62.5	0.062	0.562	addition
chlorhexidine	0.98	0.48	0.5		
CEO	1000	250	0.25	0.5	synergism
silver nitrate	7.81	1.95	0.25		
CEO	1000	250	0.25	0.5	synergism
triclosan	15.62	3.91	0.25		

**Table 9 molecules-28-00215-t009:** MICs and FIC indices of EUG alone and in combination with selected antimycotics on reference *C*. *albicans* ATCC 10231 strain.

Antifungal Agent	MIC of Antifungal Agent (µg/mL) against *C*. *albicans*		FIC	Ʃ FIC (FICI)	Interpretation
	Alone	Combination			
EUG	1000	250	0.25	1.25	indifference
nystatin	0.48	0.48	1		
EUG	1000	250	0.25	0.5	synergism
cetylpyridinium	3.91	0.98	0.25		
EUG	1000	250	0.25	0.5	synergism
chlorhexidine	7.81	1.95	0.25		
EUG	1000	250	0.25	0.75	addition
silver nitrate	7.81	3.91	0.5		
EUG	1000	250	0.25	0.75	addition
triclosan	7.81	3.91	0.5		

**Table 10 molecules-28-00215-t010:** MICs and FIC indices of EUG alone and in combination with selected antimycotics on reference *C*. *glabrata* ATCC 90030 strain.

Antifungal Agent	MIC of Antifungal Agent (µg/mL) against *C*. *glabrata*		FIC	Ʃ FIC (FICI)	Interpretation
	Alone	Combination			
EUG	1000	250	0.25	1.25	indifference
nystatin	0.48	0.48	1		
EUG	1000	250	0.25	0.5	synergism
cetylpyridinium	0.98	0.24	0.25		
EUG	1000	250	0.25	0.5	synergism
chlorhexidine	7.81	1.95	0.25		
EUG	1000	250	0.25	0.75	addition
silver nitrate	7.81	3.91	0.5		
EUG	1000	250	0.25	0.5	synergism
triclosan	15.62	3.91	0.25		

**Table 11 molecules-28-00215-t011:** MICs and FIC indices of EUG alone and in combination with selected antimycotics on reference *C*. *krusei* ATCC 14243 strain.

Antifungal Agent	MIC of Antifungal Agent (µg/mL) against *C. krusei*		FIC	Ʃ FIC (FICI)	Interpretation
	Alone	Combination			
EUG	1000	500	0.5	1.5	indifference
nystatin	0.98	0.98	1		
EUG	1000	250	0.25	0.375	synergism
cetylpyridinium	0.98	0.12	0.125		
EUG	1000	125	0.125	0.375	synergism
chlorhexidine	1.95	0.48	0.25		
EUG	1000	250	0.25	0.75	addition
silver nitrate	7.81	3.91	0.5		
EUG	1000	125	0.125	0.625	addition
triclosan	15.62	7.81	0.5		

**Table 12 molecules-28-00215-t012:** MICs and FIC indices of EUG alone and in combination with selected antimycotics on reference *C*. *parapsilosis* ATCC 22019 strain.

Antifungal Agent	MIC of Antifungal Agent (µg/mL) against *C. parapsilosis*		FIC	Ʃ FIC (FICI)	Interpretation
	Alone	Combination			
EUG	1000	250	0.5	1.25	indifference
nystatin	0.48	0.48	1		
EUG	1000	250	0.25	0.375	synergism
cetylpyridinium	1.95	0.24	0.125		
EUG	1000	62.5	0.125	0.562	addition
chlorhexidine	0.98	0.48	0.25		
EUG	1000	250	0.25	0.375	synergism
silver nitrate	7.81	0.98	0.5		
EUG	1000	250	0.125	0.5	synergism
triclosan	15.62	3.91	0.5		

## Data Availability

The data presented in this study are available on request from the corresponding author. The data are not publicly available due to privacy restrictions.

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
