# Peer review of "Clove Essential Oil and Its Main Constituent, Eugenol, as Potential Natural Antifungals against Candida spp. Alone or in Combination with Other Antimycotics Due to Synergistic Interactions"

_molecules, 2022, doi:10.3390/molecules28010215_

Round 1

Reviewer 1 Report

Dear authors, congratulations for the excellent work. My only suggestion is to put an image demonstrating the mechanisms of action. In order to make it more didactic for the reader.

Author Response

Dear Reviewer 1

Thank you very much indeed for your kind comment and the time spent on the revision of our manuscript.

Dear authors, congratulations for the excellent work. My only suggestion is to put an image demonstrating the mechanisms of action. In order to make it more didactic for the reader.

Response: Thank you very much for this comment and suggestion. We of course agree with the Reviewer and have included a suggested image demonstrating the mechanisms of antifungal action of clove essential oil and eugenol in the Discussion. We hope that this work is now more clear and understandable for the reader.

On behalf of the Authors,

Anna Biernasiuk

Reviewer 2 Report

The manuscript is relevant in anti-fungal treatment but there are some corrections and suggestions to be considered.

Line 35-36: insert citation.

Line 39: “create also” change to “also create

Line 40: Check word communiny

Line 54- 57: both sentences can be merged as one.

Line 60: better to change the scientific name from Eugenia caryophyllata to Syzygium aromaticum as Eugenia caryophyllata is the synonym

Line 65- 68: Check the sentence. Rephrasing is required.

Line 63: Is the 100 compounds from Eugenia caryophyllata or from different species of Eugenia.

Line 73-82: use individual manuscript for reviewing pharmacological effects of CEOs and EUG instead of a single review paper (reference number 5)

Table 1: Specify the Retention Index or Kovats index of each compound

Include the GC/MS chromatogram of CEOs

Expansion of MIC and MFC should be included. All tables and figures should be self-explanatory.

Why the study hasn’t done species-level identification of isolated non-albicans Candida species?

The manuscript can include a section on toxicity studies with oral cell lines as the work ultimately aims to develop an antifungal agent against Candida spp.

Author Response

Dear Reviewer 2

Thank you very much indeed for your kind comments and the time spent on the revision of our manuscript. Here below we present a point-by-point list of the responses to the obtained comments, in red.

The manuscript is relevant in anti-fungal treatment but there are some corrections and suggestions to be considered.

Response: We would like to thank you kindly for this comment and we agree with all suggestions.

Line 35-36: insert citation.

Response: Thank you for this comment. The relevant citation has been inserted.

We change to: „The predominant cause of all types of candidiases is Candida albicans – simultaneously the fourth most common etiological factor of hospital-acquired infections. It is associated with mortality rates as high as 35 – 50% [1,2].”

Line 39: “create also” change to “also create

Response: Thank you for this comment. The correction was introduced to the manuscript.

We change to: “However, superficial candidiasis, including oropharyngeal candidiasis also create an important problem due to several high risk population in the community, e.g. oncological patients”.

Line 40: Check word communiny.

Response: Thank you for this comment. The correction was introduced to the manuscript.

We change to: “However, superficial candidiasis, including oropharyngeal candidiasis also create an important problem due to several high risk population in the community, e.g. oncological patients”.

Line 54- 57: both sentences can be merged as one.

Response: Thank you for this comment. The correction was introduced to the manuscript.

The both sentences:

„They are rich mixtures of chemical compounds belonging to different chemical families, including terpenes, aldehydes, phenols, ethers, alcohols, esters, and ketones. Most of them are composed of terpenes, terpenoids, and other aromatic and aliphatic constituents with low molecular weights [3].”

We change as one to:

“They are rich mixtures of chemical compounds belonging to different chemical families, including terpenes, terpenoids, aldehydes, phenols, alcohols, ethers, esters, ketones, and other aromatic and aliphatic constituents with low molecular weights [3].”

Line 60: better to change the scientific name from Eugenia caryophyllata to Syzygium aromaticum as Eugenia caryophyllata is the synonym.

Response: Thank you for this comment. The correction was introduced to the manuscript. The scientific name of Eugenia caryophyllata was changed to Syzygium aromaticum. The nomenclature was unified.

We change to: “Clove EO (CEO) is obtained from spicy clove (Syzygium aromaticum Merr. Et Perry).”

Line 65- 68: Check the sentence. Rephrasing is required.

Response: Thank you for this comment. The correction was introduced to the manuscript. The sentence has been checked and rephrased.

The sentence: „There are also eugenol acetate, β-caryophyllene, α-ilangene, and δ-cadinene, as well as compounds with an aromatic structure, i.e. methyl eugenol, anetol, chavikol, vanillin, benzyl alcohol, cinnamic aldehyde, benzyl salicylate, and calamenene [4,5].”

We change to: „This EO also contains eugenol acetate, β-caryophyllene, α-ilangene, δ-cadinene, as well as compounds with an aromatic structure, i.e. methyl eugenol, anetol, chavikol, vanillin, benzyl alcohol, cinnamic aldehyde, benzyl salicylate, and calamenene [4,5].”

Line 63: Is the 100 compounds from Eugenia caryophyllata or from different species of Eugenia.

Response: Thank you for this comment. The correction was introduced to the manuscript. According to Wińska et al. „At present, about 100 compounds that are components of this EO have been identified”.  

The sentence: „At present, about 100 compounds that are components of this EO have been identified.”

We change to: „At present, about 100 compounds have been identified in various CEOs”.

Line 73-82: use individual manuscript for reviewing pharmacological effects of CEOs and EUG instead of a single review paper (reference number 5)

Response: Thank you for this comment. The correction was introduced to the manuscript. According to the Reviewer’s recommendation, individual manuscript was used to review the pharmacological effects of CEO and EUG instead of a single review paper (reference number 5).

Table 1: Specify the Retention Index or Kovats index of each compound.

Response: Thank you for this comment. The correction was introduced to the manuscript. The Retention Index for each compound was determined and listed in Table 1.

Include the GC/MS chromatogram of CEOs

Response: Thank you for this comment. The correction was introduced to the manuscript. The GC/MS chromatogram of CEO was presented as Figure 2.

Expansion of MIC and MFC should be included. All tables and figures should be self-explanatory.

Response: Thank you for this comment. The correction was introduced to the manuscript. The MIC and MFC extensions have been included. We hope that the tables and figures are more understandable.

Why the study hasn’t done species-level identification of isolated non-albicans Candida species?

Response: Thank you for this comment. Identification of all non-albicans Candida (NAC) species was carried out using ID 32 C tests (bioMèrieux, France). Among 30 non-albicans Candida isolates: C. tropicalis (3 isolates), C. glabrata (13 isolates), C. famata (3 isolates), C. parapsilosis (2 isolates), C. krusei (6 isolates), C. guilliermondii (2 isolates), and C. lusitaniae (1 isolates) were used (this information was included in Section 5.1.2. Microorganisms). The activity of CEO and EUG against all species identified was similar. Therefore, sensitivity to CEO and EUG was not analyzed at the species level.

In the section 5.1.2. Microorganisms:

The sentence: „The isolates were identified by standard diagnostic methods: microscopic, macroscopic, and biochemical microtest (bioMèrieux, France) on the basis of assimilation of various substrates [38].”

We change to: “The isolates were identified by standard diagnostic methods: microscopic, macroscopic, and biochemical microtest, e.g. ID 32 C (bioMèrieux, France) on the basis of assimilation of various substrates [38].”

The manuscript can include a section on toxicity studies with oral cell lines as the work ultimately aims to develop an antifungal agent against Candida spp.

Response: Thank you for this comment. We agree with the Reviewer. CEO and EUG toxicity studies on oral cell lines (both normal and cancer) will be performed during further studies.

We have already developed two antifungal preparations against Candida spp. from the oral cavity (which resulted in two patents) and we realize that such research is necessary.

We would like to kindly thank the Reviewer again for his critical comments. We agree with the Reviewer in all respects. We hope that all the mentioned shortcomings regarding the our manuscript were included in the article.

On behalf of the Authors,

Anna Biernasiuk